# COMPETITIVE EXPERIENCE REPLAY

**Hao Liu,**[*] **Alexander Trott, Richard Socher, Caiming Xiong**
Salesforce Research
Palo Alto, 94301
`lhao499@gmail.com,{atrott, rsocher, cxiong}@salesforce.com`

## ABSTRACT

Deep learning has achieved remarkable successes in solving challenging reinforcement learning (RL) problems when dense reward function is provided. However, in sparse reward environment it still often suffers from the need to carefully shape reward function to guide policy optimization. This limits the applicability of RL in the real world since both reinforcement learning and domain-specific knowledge are required. It is therefore of great practical importance to develop algorithms which can learn from a binary signal indicating successful task completion or other unshaped, sparse reward signals. We propose a novel method called competitive experience replay, which efficiently supplements a sparse reward by placing learning in the context of an exploration competition between a pair of agents. Our method complements the recently proposed hindsight experience replay (HER) by inducing an automatic exploratory curriculum. We evaluate our approach on the tasks of reaching various goal locations in an ant maze and manipulating objects with a robotic arm. Each task provides only binary rewards indicating whether or not the goal is achieved. Our method asymmetrically augments these sparse rewards for a pair of agents each learning the same task, creating a competitive game designed to drive exploration. Extensive experiments demonstrate that this method leads to faster converge and improved task performance.

## 1 INTRODUCTION

Recent progress in deep reinforcement learning has achieved very impressive results in domains ranging from playing games (Mnih et al., 2015; Silver et al., 2016; 2017; OpenAI, 2018), to high dimensional continuous control (Schulman et al., 2016; 2017; 2015), and robotics (Levine et al., 2018; Kalashnikov et al., 2018; Andrychowicz et al., 2018b).

Despite these successes, in robotics control and many other areas, deep reinforcement learning still suffers from the need to engineer a proper reward function to guide policy optimization (see e.g. Popov et al. 2017, Ng et al. 2006). In robotic control as stacking bricks, reward function need to be sharped to very complex which consists of multiple terms(Popov et al., 2017). It is extremely hard and not applicable to engineer such reward function for each task in real world since both reinforcement learning expertise and domain-specific knowledge are required. Learning to perform well in environments with sparse rewards remains a major challenge. Therefore, it is of great practical importance to develop algorithms which can learn from binary signal indicating successful task completion or other unshaped reward signal.

In environments where dense reward function is not available, only a small fraction of the agents' experiences will be useful to compute gradient to optimize policy, leading to substantial high sample complexity. Providing agents with useful signals to pursue in sparse reward environments becomes crucial in these scenarios.

---

[*]This work was done when the author was an intern at Salesforce Research.

In the domain of goal-directed RL, the recently proposed hindsight experience replay (HER) (Andrychowicz et al., 2017) addresses the challenge of learning from sparse rewards by re-labelling visited states as goal states during training. However, this technique continues to suffer from sample inefficiency, ostensibly due to difficulties related to exploration. In this work, we address these limitations by introducing a method called *Competitive Experience Replay (CER)*. This technique attempts to emphasize exploration by introducing a competition between two agents attempting to learn the same task. Intuitively, agent $A$ (the agent ultimately used for evaluation) receives a penalty for visiting states that the competitor agent ($B$) also visits; and $B$ is rewarded for visiting states found by $A$. Our approach maintains the reward from the original task such that exploration is biased towards the behaviors best suited to accomplishing the task goals. We show that this competition between agents can automatically generate a curriculum of exploration and shape otherwise sparse reward. We jointly train both agents' policies by adopting methods from multi-agent RL. In addition, we propose two versions of CER, *independent CER*, and *interact CER*, which differ in the state initialization of agent $B$: whether it is sampled from the initial state distribution or sampled from off-policy samples of agent $A$, respectively.

Whereas HER re-labels samples based on an agent's individual rollout, our method re-labels samples based on intra-agent behavior; as such, the two methods do not interfere with each other algorithmically and are easily combined during training. We evaluate our method both with and without HER on a variety of reinforcement learning tasks, including navigating an ant agent to reach a goal position and manipulating objects with a robotic arm. For each such task the default reward is sparse, corresponding to a binary indicator of goal completion. Ablation studies show that our method is important for achieving a high success rate and often demonstrates faster convergence. Interestingly, we find that CER and HER are complementary methods and employ both to reach peak efficiency and performance. Furthermore, we observe that, when combined with HER, CER outperforms curiosity-driven exploration.

## 2 BACKGROUND

Here, we provide an introduction to the relevant concepts for reinforcement learning with sparse reward (Section 2.1), Deep Deterministic Policy Gradient, the backbone algorithm we build off of, (Section 2.2), and Hindsight Experience Replay (Section 2.3).

### 2.1 SPARSE REWARD REINFORCEMENT LEARNING

Reinforcement learning considers the problem of finding an optimal policy for an agent that interacts with an uncertain environment and collects reward per action. The goal of the agent is to maximize its cumulative reward. Formally, this problem can be viewed as a Markov decision process over the environment states $s \in S$ and agent actions $a \in A$, with the (unknown) environment dynamics defined by the transition probability $T(s'|s,a)$ and reward function $r(s_t, a_t)$, which yields a reward immediately following the action $a_t$ performed in state $s_t$.

We consider goal-conditioned reinforcement learning from sparse rewards. This constitutes a modification to the reward function such that it depends on a goal $g \in G$, such that $r_g : S \times A \times G \to R$. Every episode starts with sampling a state-goal pair from some distribution $p(s_0, g)$. Unlike the state, the goal stays fixed for the whole episode. At every time step, an action is chosen according to some policy $\pi$, which is expressed as a function of the state and the goal, $\pi : S \times G \to A$. For generality, our only restriction on $G$ is that it is a subset of $S$. In other words, the goal describes a target state and the task of the agent is to reach that state. Therefore, we apply the following sparse reward function:

$$r_t = r_g(s_t, a_t, g) = \begin{cases} 0, \text{ if } |s_t - g| < \delta \\ -1, \text{ otherwise} \end{cases} \tag{1}$$

where $g$ is a goal, $|s_t - g|$ is a distance measure, and $\delta$ is a predefined threshold that controls when the goal is considered completed.

Following policy gradient methods, we model the policy as a conditional probability distribution over states, $\pi_\theta(a|[s,g])$, where $[s,g]$ denotes concatenation of state $s$ and goal $g$, and $\theta$ are the learnable parameters. Our objective is to optimize $\theta$ with respect to the expected cumulative reward, given by:

$$J(\theta) = \mathbb{E}_{s\sim\rho_\pi, a\sim\pi(a|s,g), g\sim G}\left[r_g(s,a,g)\right],$$

where $\rho_\pi(s) = \sum_{t=1}^{\infty} \gamma^{t-1}\mathrm{Pr}(s_t = s)$ is the normalized discounted state visitation distribution with discount factor $\gamma \in [0,1)$. To simplify the notation, we denote $\mathbb{E}_{s\sim\rho_\pi, a\sim\pi(a|s,g), g\sim G}[\cdot]$ by simply $\mathbb{E}_\pi[\cdot]$ in the rest of paper. According to the policy gradient theorem (Sutton et al., 1998), the gradient of $J(\theta)$ can be written as

$$\nabla_\theta J(\theta) = \mathbb{E}_\pi\left[\nabla_\theta \log \pi(a|s,g)Q^\pi(s,a,g)\right], \qquad (2)$$

where $Q^\pi(s,a,g) = \mathbb{E}_\pi\left[\sum_{t=1}^{\infty}\gamma^{t-1}r_g(s_t,a_t,g)|s_1 = s, a_1 = a\right]$, called the critic, denotes the expected return under policy $\pi$ after taking an action $a$ in state $s$, with goal $g$.

## 2.2 DDPG ALGORITHM

Here, we introduce Deep Deterministic Policy Gradient (DDPG) (Lillicrap et al., 2015), a model-free RL algorithm for continuous action spaces that serves as our backbone algorithm. Our proposed modifications need not be restricted to DDPG; however, we leave experimentation with other continuous control algorithms to future work. In DDPG, we maintain a deterministic target policy $\mu(s,g)$ and a critic $Q(s,a,g)$, both implemented as deep neural networks (note: we modify the standard notation to accommodate goal-conditioned tasks). To train these networks, episodes are generated by sampling actions from the policy plus some noise, $a \sim \mu(s,g) + N(0,1)$. The transition tuple associated with each action $(s_t, a_t, g_t, r_t, s_{t+1})$ is stored in the so-called replay buffer. During training, transition tuples are sampled from the buffer to perform mini-batch gradient descent on the loss $L$ which encourages the approximated Q-function to satisfy the Bellman equation $L = \mathbb{E}(Q(s_t, a_t, g_t) - y_t)^2$, where $y_t = r_t + \gamma Q(s_{t+1}, \mu(s_{t+1}, g_t), g_t)$. Similarly, the actor can be updated by training with mini-batch gradient descent on the loss $J(\theta) = -\mathbb{E}_s Q(s, \mu(s,g), g)$ through the deterministic policy gradient algorithm (Silver et al., 2014),

$$\nabla_\theta J(\theta) = \mathbb{E}[\nabla_\theta\mu(s,g)\nabla_a Q(s,a,g)|_{a=\mu(s,g)}]. \qquad (3)$$

To make training more stable, the targets $y_t$ are typically computed using a separate target network, whose weights are periodically updated to the current weights of the main network (Lillicrap et al., 2015; Mnih et al., 2013; 2015).

## 2.3 HINDSIGHT EXPERIENCE REPLAY

Despite numerous advances in the application of deep learning to RL challenges, learning in the presence of sparse rewards remains a major challenge. Intuitively, these algorithms depend on sufficient variability within the encountered rewards and, in many cases, random exploration is unlikely to uncover this variability if goals are difficult to reach. Recently, Andrychowicz et al. (2017) proposed Hindsight Experience Replay (HER) as a technique to address this challenge. The key insight of HER is that failed rollouts (where no task reward was obtained) can be treated as successful by assuming that a visited state was the actual goal. Basically, HER amounts to a relabelling strategy. For every episode the agent experiences, it gets stored in the replay buffer twice: once with the original goal pursued in the episode and once with the goal replaced with a future state achieved in the episode, as if the agent were instructed to reach this state from the beginning. Formally, HER randomly samples a mini-batch of episodes in buffer, for each episode $(\{s^i\}_{i=1}^T, \{g^i\}_{i=1}^T, \{a^i\}_{i=1}^T, \{r^i\}_{i=1}^T, \{s'^i\}_{i=1}^T)$, for each state $s_t$, where $1 \le t \le T-1$ in an episode, we

randomly choose $s_k$ where $t + 1 \leq k \leq T$ and relabel transition $(s_t, a_t, g_t, r_t, s_{t+1})$ to $(s_t, a_t, s_k, r'_t, s_{t+1})$ and recalculate reward $r'_t$,

$$r'_t = r_g(s_t, s_k) = \begin{cases} 0, \text{ if } |s_t - s_k| < \delta, \\ -1, \text{ otherwise.} \end{cases} \tag{4}$$

## 3 METHOD

In this section, we present Competitive Experience Replay (CER) for policy gradient methods (Section 3.1) and describe the application of multi-agent DDPG to enable this technique (Section 3.2).

### 3.1 COMPETITIVE EXPERIENCE REPLAY

While the re-labelling strategy introduced by HER provides useful rewards for training a goal-conditioned policy, it assumes that learning from arbitrary goals will generalize to the actual task goals. As such, exploration remains a fundamental challenge for goal-directed RL with sparse reward. We propose a re-labelling strategy designed to overcome this challenge. Our method is partly inspired by the success of self-play in learning to play competitive games, where sparse rewards (i.e. win or lose) are common. Rather than train a single agent, we train a pair of agents on the same task and apply an asymmetric reward re-labelling strategy to induce a competition designed to encourage exploration. We call this strategy *Competitive Experience Replay* (CER).

To implement CER, we learn a policy for each agent, $\pi_A$ and $\pi_B$, as well as a multi-agent critic (see below), taking advantage of methods for decentralized execution and centralized training. During decentralized execution, $\pi_A$ and $\pi_B$ collect DDPG episode rollouts in parallel. Each agent effectively plays a single-player game; however, to take advantage of multi-agent training methods, we arbitrarily pair the rollout from $\pi_A$ with that from $\pi_B$ and store them as a single, multi-agent rollout in the replay buffer $\mathcal{D}$. When training on a mini-batch of off policy samples, we first randomly sample a mini-batch of episodes in $\mathcal{D}$ and then randomly sample transitions in each episode. We denote the resulting mini-batch of transitions as $\{(s_A^i, a_A^i, g_A^i, r_A^i, s'^i_A), (s_B^i, a_B^i, g_B^i, r_B^i, s'^i_B)\}_{i=1}^m$, where $m$ is the size of the mini-batch.

Reward re-labelling in CER attempts to create an implicit exploration curriculum by punishing agent $A$ for visiting states that agent $B$ also visited, and, for those same states, rewarding agent $B$. For each $A$ state $s_A^i$ in a mini-batch of transitions, we check if any $B$ state $s_B^j$ in the mini-batch satisfies $|s_A^i - s_B^j| < \delta$ and, if so, re-label $r_A^i$ with $r_A^i - 1$. Conversely, each time such a nearby state pair is found, we increment the associated reward for agent $B$, $r_B^j$, by $+1$. Each transition for agent $A$ can therefore be penalized only once, whereas no restriction is placed on the extra reward given to a transition for agent $B$. Following training both agents with the re-labelled rewards, we retain the policy $\pi_A$ for evaluation. Additional implementation details are provided in the appendix (Section D).

We focus on two variations of CER that satisfy the multi-agent self-play requirements: first, the policy $\pi_B$ receives its initial state from the task's initial state distribution; second, although more restricted to re-settable environments, $\pi_B$ receives its initial state from a random off-policy sample of $\pi_A$. We refer to the above methods as *independent-CER* and *interact-CER*, respectively, in the following sections.

Importantly, CER re-labels rewards based on intra-agent behavior, whereas HER re-labels rewards based on each individual agent's behavior. As a result, the two methods can be easily combined. In fact, as our experiments demonstrate, CER and HER are complementary and likely reflect distinct challenges that are both addressed through reward re-labelling.

## 3.2 GOAL CONDITIONED MULTI-AGENT LEARNING

We extend multi-agent DDPG (MADDPG), proposed by Lowe et al. (2017), for training using CER. MAD-DPG attempts to learn a different policy per agent and a single, centralized critic that has access to the combined states, actions, and goals of all agents.

More precisely, consider a game with $N$ agents with policies parameterized by $\boldsymbol{\theta} = \{\theta_1, \ldots, \theta_N\}$, and let $\boldsymbol{\pi} = \{\pi_1, \ldots, \pi_N\}$ be the set of all agent policies. $\mathbf{g} = [g_1, \ldots, g_N]$ represents the concatenation of each agent's goal, $\mathbf{s} = [s_1, \ldots, s_N]$ the concatenated states, and $\mathbf{a} = [a_1, \ldots, a_N]$ the concatenated actions. With this notation, we can write the gradient of the expected return for agent $i$, $J(\theta_i) = \mathbb{E}[R_i]$ as:

$$\nabla_{\theta_i} J(\theta_i) = \mathbb{E}_{\boldsymbol{\pi}}[\nabla_{\theta_i} \log \pi_i(a_i|s_i, g_i) Q_i^{\boldsymbol{\pi}}(\mathbf{s}, \mathbf{a}, \mathbf{g})]. \tag{5}$$

With deterministic policies $\boldsymbol{\mu} = \{\mu_1, \ldots, \mu_N\}$, the gradient becomes:

$$\nabla_{\theta_i} J(\theta_i) = \mathbb{E}_{\boldsymbol{\mu}}[\nabla_{\theta_i} \mu_i(s_i, g_i) \nabla_{a_i} Q_i^{\boldsymbol{\mu}}(\mathbf{s}, \mathbf{a}, \mathbf{g})|_{a_i = \mu_i(s_i, g_i)}], \tag{6}$$

The centralized action-value function $Q_i^{\boldsymbol{\mu}}$, which estimates the expected return for agent $i$, is updated as:

$$\mathcal{L}(\theta_i) = \mathbb{E}_{\mathbf{s}, \mathbf{a}, \mathbf{r}, \mathbf{s}'}[(Q_i^{\boldsymbol{\mu}}(\mathbf{s}, \mathbf{a}, \mathbf{g}) - y)^2], \quad y = r_i + \gamma\, Q_i^{\boldsymbol{\mu}'}(\mathbf{s}', a_1', \ldots, a_N', \mathbf{g})\big|_{a_j' = \boldsymbol{\mu}_j'(s_j)}, \tag{7}$$

where $\boldsymbol{\mu}' = \{\boldsymbol{\mu}_{\theta_1'}, \ldots, \boldsymbol{\mu}_{\theta_N'}\}$ is the set of target policies with delayed parameters $\theta_i'$. In practice people usually soft update it as $\theta_i' \leftarrow \tau \theta_i + (1 - \tau)\theta_i'$, where $\tau$ is a Polyak coefficient.

During training, we collect paired rollouts as described above, apply any re-labelling strategies (such as CER or HER) and use the MADDPG algorithm to train both agent policies and the centralized critic, concatenating states, actions, and goals where appropriate. Putting everything together, we summarize the full method in Algorithm 1.

## 4 EXPERIMENT

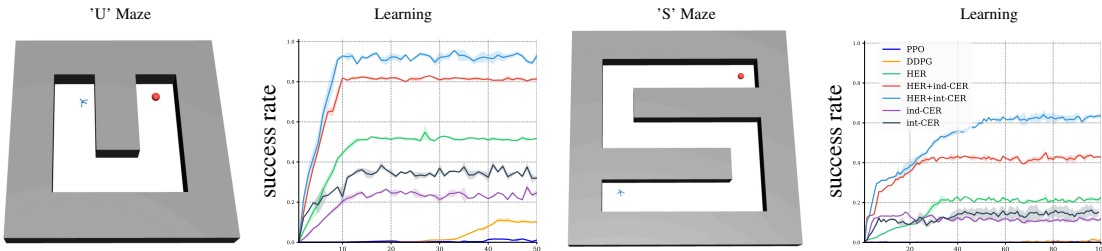

Figure 1: Goal conditioned 'U' shaped maze and 'S' shaped maze. For each plot pair, the left illustrates a rendering of the maze type and the right shows the associated success rate throughout training for each method. Each line shows results averaged over 5 random initializations; the X-axis corresponds to epoch number and shaded regions show standard deviation. All metrics based on CER reference the performance of agent $A$.

### 4.1 COMBINING COMPETITIVE EXPERIENCE REPLAY WITH EXISTING METHODS

We start by asking whether CER improves performance and sample efficiency in a sparse reward task. To this end, we constructed two different mazes, an easier 'U' shaped maze and a more difficult 'S' shaped maze (Figure 1). The goal of the ant agent is to reach the target mark by a red sphere, whose location is randomly sampled for each new episode. At each step, the agent obtains a reward of 0 if the goal has been achieved and

−1 otherwise. Additional details of the ant maze environments are found in Appendix B. An advantage of this environment is that it can be reset to any given state, facilitating comparison between our two proposed variants of CER (*int*-CER requires the environment to have this feature).

We compare agents trained with HER, both variants of CER, and both variants of CER with HER. Since each uses DDPG as a backbone algorithm, we also include results from a policy trained using DDPG alone. To confirm that any difficulties are not simply due to DDPG, we include results from a policy trained using Proximal Policy Optimization (PPO) (Schulman et al., 2017). The results for each maze are shown in Figure 1. DDPG and PPO each performs quite poorly by itself, likely due to the sparsity of the reward in this task set up. Adding CER to DDPG partially overcomes this limitation in terms of final success rate, and, notably, reaches this stronger result with many fewer examples. A similar result is seen when adding HER to DDPG. Importantly, adding both HER and CER offers the best improvement of final success rate without requiring any observable increase in the number of episodes, as compared to each of the other baselines. These results support our hypothesis that existing state-of-the-art methods do not sufficiently address the exploration challenge intrinsic to sparse reward environments. Furthermore, these results show that CER improves both the quality and efficiency of learning in such challenging settings, especially when combined with HER. These results also show that *int*-CER tends to outperform *ind*-CER. As such, *int*-CER is considered preferable but has more restrictive technical requirements.

To examine the efficacy of our method on a broader range of tasks, we evaluate the change in performance when *ind*-CER is added to HER on the challenging multi-goal sparse reward environments introduced in Plappert et al. (2018). (Note: we would prefer to examine *int*-CER but are prevented by technical issues related to the environment.) Results for each of the 12 tasks we trained on are illustrated in Figure 3. Our method, when used on top of HER, improves performance wherever it is not already saturated. This is especially true on harder tasks, where HER alone achieves only modest success (e.g., HandManipulateEggFull and handManipulatePenRotate). These results further support the conclusion that existing methods often fail to achieve sufficient exploration. Our method, which provides a targeted solution to this challenge, naturally complements HER.

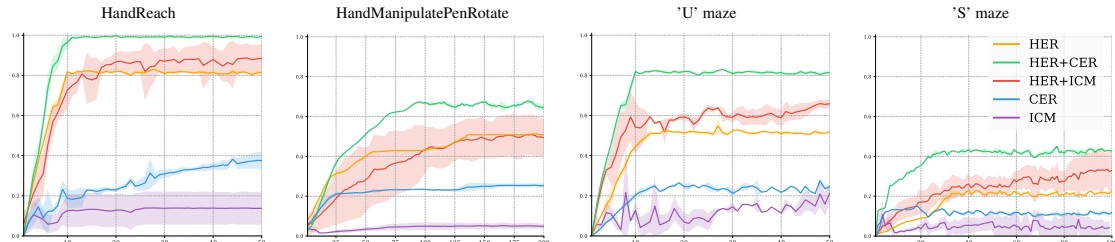

Figure 2: Comparing CER and ICM with and without HER. Each line shows task success rate averaged over 5 random initializations. X-axis is epoch number; shaded regions denote standard deviation.

## 4.2 COMPARING CER WITH CURIOSITY-DRIVEN EXPLORATION

CER is designed to encourage exploration, but, unlike other methods, uses the behavior of a competitor agent to automatically determine the criteria for exploratory reward. Numerous methods have been proposed for improving exploration; for example, count-based exploration (Bellemare et al., 2016; Tang et al., 2017), VIME (Houthooft et al., 2016), bootstrap DQN (Osband et al., 2016), goal exploration process (Forestier et al., 2017), parameter space noise (Plappert et al., 2017), dynamics curiosity and boredom (Schmidhuber, 1991), and EX2 (Fu et al., 2017). One recent and popular method, intrinsic curiosity module (ICM) (Pathak et al., 2017; Burda et al., 2018), augments task reward with curiosity-driven reward. Much like CER and HER, ICM provides a method to relabel the reward associated with each transition; this reward comes from the

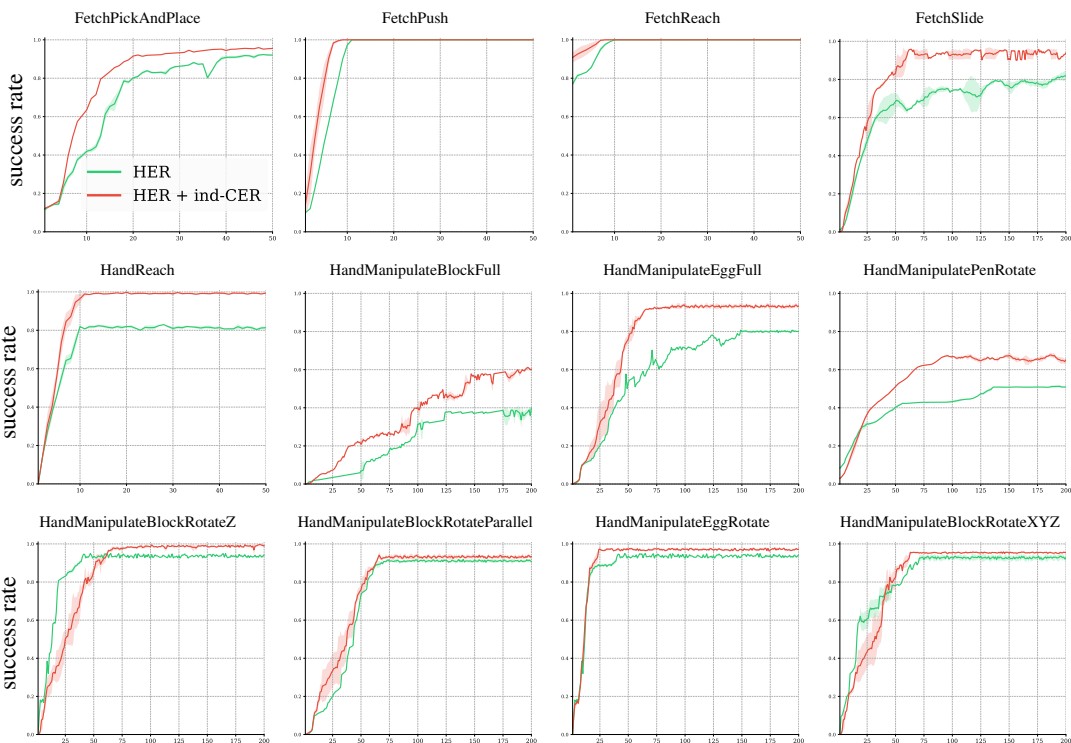

Figure 3: Evaluation of HER with *ind*-CER across different robotic control environments. Each line shows results averaged over 5 random initializations. X-axis shows epoch number; shaded regions denote standard deviation.

error in a jointly trained forward prediction model. We compare how CER and ICM affect task performance and how they interact with HER across 4 tasks where we were able to implement ICM successfully. Figure 2 shows the results on several robotic control and maze tasks. We observe CER to consistently outperform ICM when each is implemented in isolation. In addition, we observe HER to benefit more from the addition of CER than of ICM. Compared to ICM, CER also tends to improve the speed of learning. These results suggest that the underlying problem is not a lack of diversity of states being visited but the size of the state space that the agent must explore (as also discussed in Andrychowicz et al. (2017)). One interpretation is that, since the two CER agents are both learning the task, the dynamics of their competition encourage more task-relevant exploration. From this perspective, CER provides an automatic curriculum for exploration such that visited states are utilized more efficiently.

### 4.3 ANALYSIS OF CER

Figure 4 illustrates the success rates of agents $A$ and $B$ as well as the 'effect ratio,' which is the fraction of mini-batch samples whose reward is changed by CER during training, calculated as $\phi = \frac{N}{M}$ where $N$ is the number of samples changed by CER and $M$ is the number of samples in mini-batch. We observe a strong correspondence between the success rate and effect ratio, likely reflecting the influence of CER on the learning dynamics. While a deeper analysis would be required to concretely understand the interplay of these two terms, we point out that CER re-labels a substantial fraction of rewards in the mini-batch. Interestingly, even the relatively small effect ratio observed in the first few epochs is enough support rapid learning. We speculate that the sampling strategy used in *int*-CER provides a more targeted re-labelling, leading to the more rapid increase in success rate for agent $A$. We observe that agent $B$ reaches a lower level of performance.

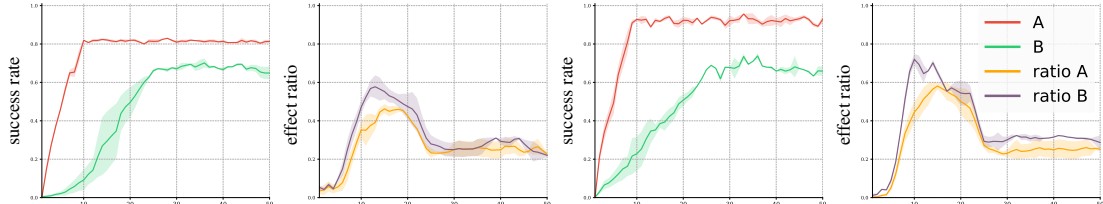

Figure 4: Illustration of the automatically generated curriculum between $A$ and $B$ on 'U' shaped AntMaze task. From left to right are success rate of *ind-CER*, effect ratio of *ind-CER*, success rate of *int-CER*, and effect ratio of *int-CER*. Each line shows results averaged over 5 random initializations. X-axis is epoch number.

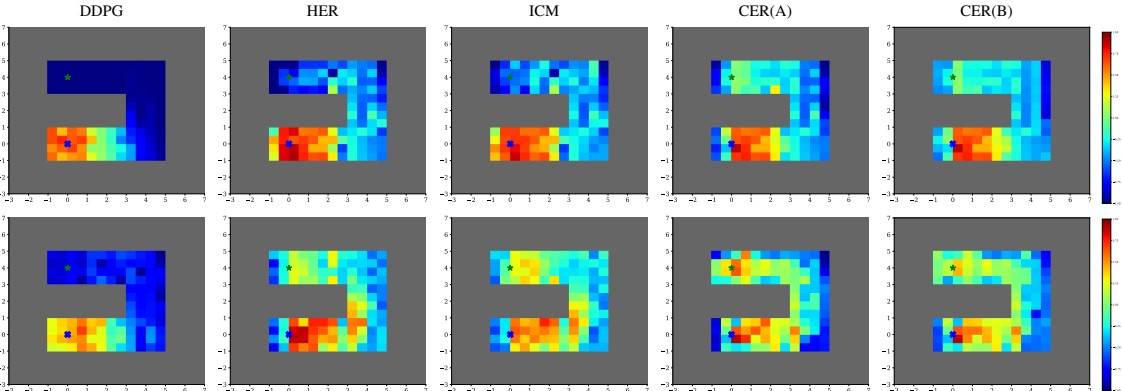

Figure 5: State visitation frequency on 'U' maze. Top row is from after 5 epochs; bottom row from after 35. Blue on the colormap denotes no visitation. The start and goal locations are labeled with the blue and green markers, respectively. ICM and CER are trained with HER also.

This likely results from resetting the parameters of agent $B$ periodically during early training, which we observe to improve the ultimate performance of agent $A$ (see Section D for details). It is also possible that the reward structure of CER asymmetrically benefits agent $A$ with respect to the underlying task.

To gain additional insight into how each method influences the behaviors learned across training, we visualize the frequency with which each location in the 'U' maze is visited after the 5th and 35th training epoch (Figure 5). Comparing DDPG and HER shows that the latter clearly helps move the state distribution towards the goal, especially during the early parts of training. When CER is added, states near the goal create a clear mode in the visitation distribution. This is never apparent with DDPG alone and only obvious for HER and ICM+HER later in training. CER also appears to focus the visitation of the agent, as it less frequently gets caught along the outer walls. These disparities are emphasized in Figure 6, where we show the difference in the visitation profiles. The left two plots compare CER+HER vs. HER. The right two plots compare Agent A vs Agent B from CER+HER. Interestingly, the Agents A and B exhibit fairly similar aggregate visitation profiles with the exception that Agent A reaches the goal more often later during training. These visitation profiles underscore both the quantitative and qualitative improvements associated with CER.

## 5 RELATED WORK

Self-play has a long history in this domain of research (Samuel, 1959; Tesauro, 1995; Heess et al., 2017). Silver et al. (2017) use self-play with deep reinforcement learning techniques to master the game of Go; and self-play has even been applied in Dota 5v5 (OpenAI, 2018).

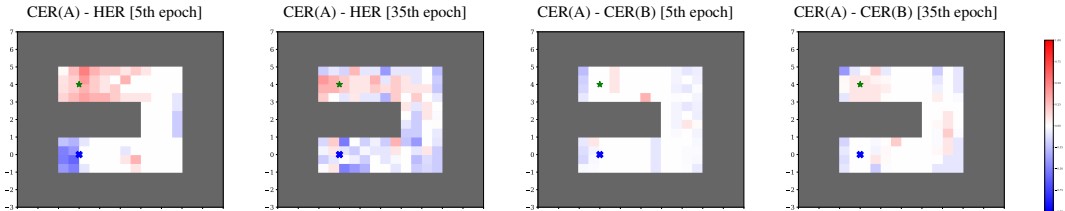

Figure 6: State visitation difference on 'U' maze. Left plots compare CER against HER; right plots compare the two CER agents. Each plot illustrates the difference in visitation, with red indicating more visitation by CER(A). Note: here, the "CER" agent is trained with CER+HER.

Curriculum learning is widely used for training neural networks (see e.g., Bengio et al., 2009; Graves et al., 2017; Elman, 1993; Olsson, 1995). A general framework for automatic task selection is Powerplay (Schmidhuber, 2013; Srivastava et al., 2013), which proposes an asymptotically optimal way of selecting tasks from a set of tasks with program search, and use replay buffer to avoid catastrophic forgetting. Florensa et al. (2017); Held et al. (2017) propose to automatically adjust task difficulty during training. Forestier et al. (2017) study intrinsically motivated goal for robotic learning. Recently, Sukhbaatar et al. (2017) suggest to use self-play between two agents where reward depends on the time of the other agent to complete to enable implicit curriculum. Our work is similar to theirs, but we propose to use sample-based competitive experience replay, which is not only more readily scalable to high-dimension control but also integrates easily with Hindsight Experience Replay (Andrychowicz et al., 2017). The method of initializing based on a state not sampled from the initial state distribution has been explored in other works. For example, Ivanovic et al. (2018) propose to create a backwards curriculum for continuous control tasks through learning a dynamics model. Resnick et al. (2018b) and Salimans & Chen (2018) propose to train policies on Pommerman (Resnick et al., 2018a) and the Atari game 'Montezumas Revenge' by starting each episode from a different point along a demonstration. Recently, Goyal et al. (2018) and Edwards et al. (2018) propose a learned backtracking model to generate traces that lead to high value states in order to obtain higher sample efficiency.

Experience replay has been introduced in (Lin, 1992) and later was a crucial ingredient in learning to master Atari games (Mnih et al., 2013; 2015). Wang et al. (2016) propose truncation with bias correction to reduce variance from using off-policy data in buffer and achieves good performance on continuous and discrete tasks. Different approaches have been proposed to incorporate model-free learning with experience replay(Gu et al., 2016; Feng et al., 2019). Schaul et al. (2015) improve experience replay by assigning priorities to transitions in the buffer to efficiently utilize samples. Horgan et al. (2018) further improve experience replay by proposing a distributed RL system in which experiences are shared between parallel workers and accumulated into a central replay memory and prioritized replay is used to update the policy based on the diverse accumulated experiences.

## 6 CONCLUSION

We introduce Competitive Experience Replay, a new and general method for encouraging exploration through implicit curriculum learning in sparse reward settings. We demonstrate an empirical advantage of our technique when combined with existing methods in several challenging RL tasks. In future work, we aim to investigate richer ways to re-label rewards based on intra-agent samples to further harness multi-agent competition, it's interesting to investigate counterfactual inference to promote efficient re-label off-policy samples. We hope that this will facilitate the application of our method to more open-end environments with even more challenging task structures. In addition, future work will explore integrating our method into approaches more closely related to model-based learning, where adequate exposure to the dynamics of the environment is often crucial.

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

## A   ALGORITHM

We summarize the algorithm for HER with CER in Algorithm 1.

---

**Algorithm 1** HER with CER

---

Initialize a random process $\mathcal{N}$ for action exploration, max episode length to $L$
Set CER to *ind-CER* or *int-CER*
**for** episode $= 1$ to $M$ **do**
   Receive initial state $s_A$
   Receive a goal $r_A$ for this episode
   Initialize episode buffer *buffer$_A$*
   **for** $t = 1$ to $L$ **do**
      select action $a_A = \boldsymbol{\mu}_{\theta_i}(s_A) + \mathcal{N}_t$ w.r.t. the current policy and exploration
      Execute actions $a_A$ and observe reward $r_A$ and new state $s_A'$
      Store $(s_A, a_A, g_A, r_A, s_A')$ in *buffer$_A$*
      $s_A \leftarrow s_A'$
   **end for**
   Receive initial state $s_B$ or Receive initial state $s_B$, where $s_B$ is a state sampled from *buffer$_A$*
   Receive a goal $r_B$ for this episode
   Initialize episode buffer *buffer$_B$*
   **for** $t = 1$ to $L$ **do**
      select action $a_B = \boldsymbol{\mu}_{\theta_i}(s_B) + \mathcal{N}_t$ w.r.t. the current policy and exploration
      Execute actions $a_B$ and observe reward $r_B$ and new state $s_B'$
      Store $(s_B, a_B, g_B, r_B, s_B')$ in *buffer$_B$*
      $s_B \leftarrow s_B'$
   **end for**
   Concatenate *buffer$_A$* and *buffer$_B$* and store $(\{\mathbf{s^i}\}_{i=1}^T, \{\mathbf{a^i}\}_{i=1}^T, \{\mathbf{g^i}\}_{i=1}^T, \{\mathbf{r^i}\}_{i=1}^T, \{\mathbf{s'^i}\}_{i=1}^T)$ in replay buffer $\mathcal{D}$
   *// Optimization based on off-policy samples*
   **for** k $= 1$ to $K$ **do**
      *// Relabelling off-policy samples*
      Sample a random minibatch of $S$ samples $(\mathbf{s}^j, \mathbf{a}^j, \mathbf{g}^j, \mathbf{r}^j, \mathbf{s'}^j)$ from $\mathcal{D}$
      Apply HER strategy on samples
      Apply *ind-CER* or *int-CER* strategy on samples
      **for** agent $i = A, B$ **do**
         Do one step optimization based on Eq (6) and Eq (7), and update target networks.
      **end for**
   **end for**
**end for**

---

## B   ENVIRONMENT DETAILS

**Robotic control environments**   The robotic control environments shown in Figure 7 are part of challenging continuous robotics control suite (Plappert et al., 2018) integrated in OpenAI Gym (Brockman et al., 2016) and based on Mujoco simulation enginge (Todorov et al., 2012). The Fetch environments are based on the 7-DoF Fetch robotics arm, which has a two-fingered parallel gripper. The Hand environments are based on the Shadow Dexterous Hand, which is an anthropomorphic robotic hand with 24 degrees of freedom. In all tasks, rewards are sparse and binary: the agent obtains a reward of $0$ if the goal has been achieved and $-1$ otherwise. For more details please refer to Plappert et al. (2018).

**Ant maze environments**   We also construct two maze environments with the *re-settable* property, rendered in Figure 1, in which walls are placed to construct "U" or "S" shaped maze. At the beginning of each episode,

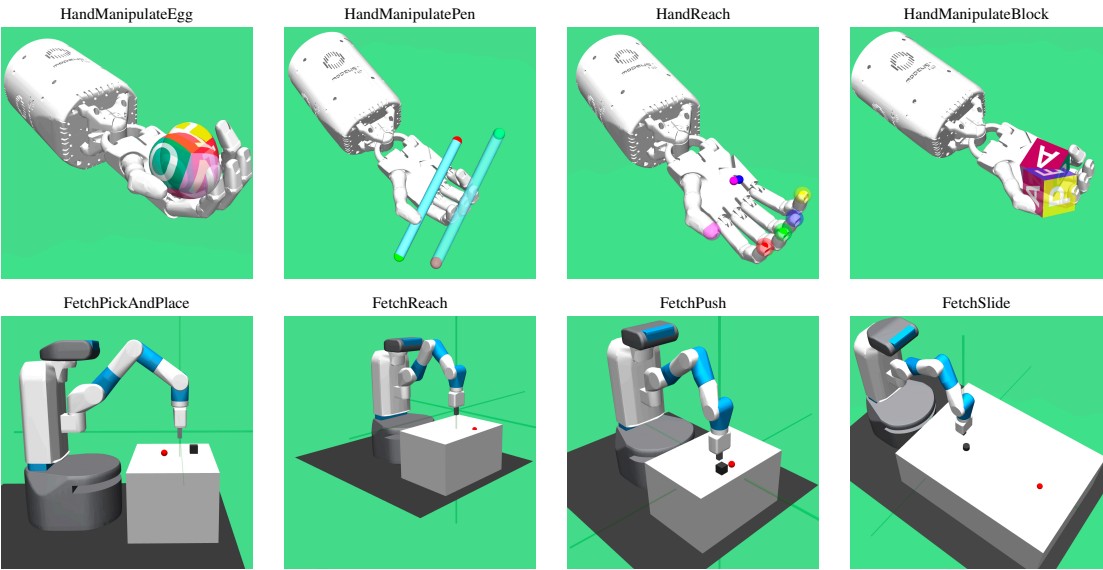

Figure 7: Robotics control environments used in our experiments

the agent is always initialized at position $(0, 0)$, and can move in any direction as long as it is not obstructed by a wall. The x- and y-axis locations of the target ball are sampled uniformly from $[-5, 20], [-5, 20]$, respectively.

## C  COMPETITIVE EXPERIENCE REPLAY WITH DIFFERENT ADVERSARIAL AGENTS

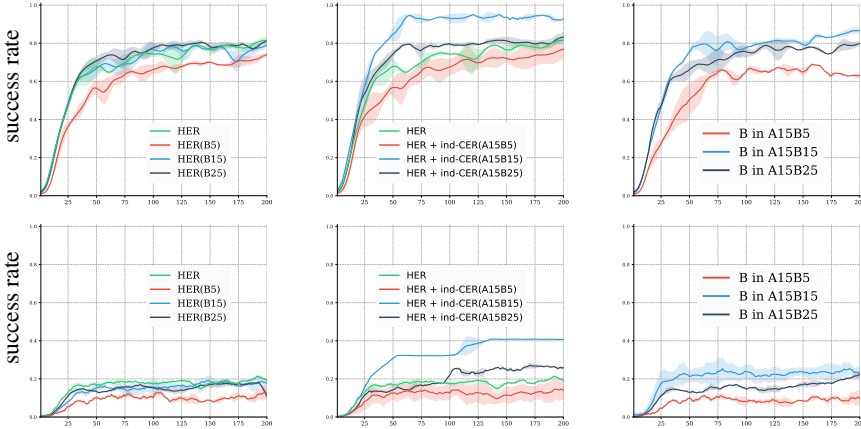

Figure 8: Performance on FetchSlide (top) and HandManipulatePenFull (bottom) for different batch-size multipliers when training using HER (left columns) or HER+CER (middle, right columns). Middle plots show the performance of agent $A$ and, in the right plots, performance of agent $B$ is shown for reference.

As Andrychowicz et al. (2017) and Andrychowicz et al. (2018a) observe (and we also observe), performance depends on the batch size. We leverage this observation to tune the relative strengths of Agents $A$ and $B$ by separately manipulating the batch sizes used for updating each.

For simplicity, we control the batch size by changing the number of MPI workers devoted to a particular update. Each MPI worker computes the gradients for a batch size of 256; averaging the gradients from each worker results in an effective batch size of $N * 256$. For our single-agent baselines, we choose $N = 30$ workers, and, when using CER, a default of $N = 15$ for each $A$ and $B$ In the following, $AxBy$ denotes, for agent $A$, $N = x$ and, for agent $B$, $N = y$. These results suggest that, while a sufficiently large batch size is important for achieving the best performance, the optimal configuration occurs when the batch sizes used for the two agents are balanced. Interestingly, we observe that batch size imbalance adversely effects both agents trained during CER.

## D    Implementation details

In the code for HER (Andrychowicz et al., 2017), the authors use MPI to increase the batch size. MPI is used here to run rollouts in parallel and average gradients over all MPI workers. We found that MPI is crucial for good performance, since training for longer time with a smaller batch size gives sub-optimal performance. This is consistent with the authors' findings in their code that having a much larger batch size helps a lot. For each experiment, we provide the per-worker batch sizes in Table 1; note that the effective batch size is multiplied by the number of MPI workers $N$.

In our implementation, we do $N$ rollouts in parallel for each agent and have separate optimizers for each. We found that periodically resetting the parameters of agent $B$ in early stage of training helps agent $A$ more consistently reach a high level of performance. This resetting helps to strike an optimal balance between the influences of HER and CER in training agent $A$. We also add L2 regularization, following the practice of Andrychowicz et al. (2017).

For the experiments with neural networks, all parameters are randomly initialized from $\mathcal{N}(0, 0.2)$. We use networks with three layers of hidden layer size 256 and Adam (Kingma & Ba, 2014) for optimization. Presented results are averaged over 5 random seeds. We summarize the hyperparameters in Table 1.

| Hyperparameter name | 'U' AntMaze | 'S' AntMaze | Fetch Control | Hand Control |
|---|---|---|---|---|
| Buffer size | $1E5$ | $1E6$ | $1E6$ | $1E6$ |
| Batch size | 128 | 128 | 256 | 256 |
| Max steps of episode | 50 | 100 | 50 | 100 |
| Reset epochs | 2 | 2 | 5 | 5 |
| Max reset epochs | 10 | 10 | 20 | 30 |
| Total epochs | 50 | 100 | 100/50 | 200 |
| Actor Learning rate | 0.0004 | 0.0004 | 0.001 | 0.001 |
| Critic Learning rate | 0.0004 | 0.0004 | 0.001 | 0.001 |
| Action L2 regularization | 0.01 | 0.01 | 1.00 | 1.00 |
| Polyak | 0.95 | 0.95 | 0.95 | 0.95 |

Table 1: Hyperparameter values used in experiments.

