# OpenReview forum: "Competitive experience replay"
_ICLR.cc/2019/Conference_

### Official Review · AnonReviewer3 · 2018-11-02
**clear simple idea and good results**

**Rating:** 7
**Confidence:** 5

**Review:**

The paper is well written and easy to read. Exploration is one of the fundamental problems in RL, and the idea of using two agents for better exploration is interesting and novel. However, an explanation of the intuition behind the method would be useful. The experimental results show that the method works well in complex tasks. Since states are compared to each other in L2 distance, the method might not generalize to other domains where L2 distance is not a good distance metric.

Pros:
- well written
- a simple and novel idea tackling a hard problem
- good results on hard tasks

Cons:
- an explanation of why the method should work is missing
- plot text is too small (what is the unit of X-axis?)

Questions:
- what is the intuition behind the method?
- during training, randomly sampled two states are compared. why it is a good idea? how the replay buffer size will affect it?
- since it is a two-player game, is there anything you can say about its Nash equilibrium?
- why A is better than B at the task?
- when comparing states, are whole raw observations (including velocity etc.) used?
- section 4.2 doesn't seem to be that relevant or helpful. is it really necessary?
- fig 4 is missing CER alone results? why is that? it doesn't work by itself on those tasks?

---

> ### Author Response · Authors · 2018-11-25
> **Response to Reviewer 3**
>
> Thank you for your thoughtful and constructive comments. Following the reviewer’s suggestions, we have re-written portions of the paper to hopefully provide better insights into why CER improves performance. In addition, we have taken care to improve the quality of the figures to improve readability.
> Since the policy does not include any awareness of the other agent, no fixed correspondence between rollouts is required and the randomness offers a better sampling. The replay buffer size may come in to play if it induces comparison between rollouts from very different stages of training. Even so, the agents are intentionally different and MADDPG helps to make the learning problem more stationary. As such, we expect our results to be fairly robust to the size of the replay buffer.
> Unfortunately, we are unable to ground this method or the results within such a theoretical lens at the time being. However, we would hope to explore that possibility more before our paper is finalized or in future work.
> The most likely explanation for B being worse than A is simply that, during the initial stages of training, the parameters of B are periodically re-initialized. We find that this practice improves stability during training and the consistency of Agent A’s final performance. However, there is likely also an effect due to the asymmetric reward structure, such that the reward received by Agent A is more helpful for learning the underlying task itself. In anticipation of other readers having the same question, we have made these details clearer in the paper.
> Regarding the state comparison, we apply the same criteria used to measure goal completion. So, only a small portion of the whole state is used.
> We agree that Section 4.2 is more appropriate as a section of the appendix. We have rearranged the paper accordingly.
> Based on the results earlier in the paper, the empirical best application of CER is in combination with HER. Our intention with Figure 4 was simply to provide a more exhaustive demonstration of the benefit gained by CER when added to the previous best method (HER). As discussed in the paper, we do not wish for HER and CER to be viewed as alternative methods but instead complementary methods. We use earlier figures to report the results of CER without HER.

---

### Official Review · AnonReviewer1 · 2018-11-02
**To address the sparse reward problems, the authors propose a relabeling strategy called Competitive Experience Reply (CER).  This strategy relabels states, and places learning in the context of an exploration competition between a pair of agents.  The experiments support some parts of authors’ claim well.  However, the experiments are insufficient.**

**Rating:** 6
**Confidence:** 4

**Review:**

The authors propose a states relabeling strategy (CER) to encourage exploration in RL algorithms by organizing a competitive game between a pair of agents.
To verify their strategy, they extend MADDPG as their framework. Then, they compare the performance of agents trained with HER, and both variants of CER, and both variants of CER with HER. The experiments show that CER can improve the performance of HER with faster converge and higher accuracy.

My major concerns are as follows.
1.	The authors may want to conduct more experiments to compare CER with other state-of-the-art methods such as PPO[1]. As illustrated in Figure 1, the performance of HER is better than that of CER. The authors may want to analyze whether CER strategy alone could properly address the sparse reward problems, and why CER strategy can improve HER. The authors have mentioned that CER is “orthogonal” to HER. I suggest authors provide more discussions on this statement.
2.	The authors may want to improve the readability of this paper.
For example, in Figure 1, the authors may want to clarify the meanings of the axes and the plots.
The results shown in Figure 3 are confusing. How can the authors come to the conclusion that the optimal conﬁguration requires balancing the batch sizes used for the two agents?
To better illustrate the framework of CER, the authors may want to show its flow chart.
3.	There are some typos. For example, in Section 2.1, the authors use T(s’|s,a) without index t; in Section 2.2, the authors use both Q(a,s,g) and Q(s,a,g).
There is something wrong with the format of the reference (“Tim Salimans and Richard Chen … demonstration/, 2018.”) in the bottom of page 10.

[1] Schulman J, Wolski F, Dhariwal P, et al. Proximal Policy Optimization Algorithms[J]. 2017.

---

> ### Author Response · Authors · 2018-11-25
> **Response to Reviewer 1**
>
> We thank the reviewer for their constructive and thoughtful feedback. Our updated version of the paper includes a number of changes designed to improve readability and consistency with respect to plots and notation. In addition, we have moved Figure 3 and its accompanying text to the appendix, since those results are more related to practical considerations that we wanted readers to be aware of in case they cared to implement our methods. We have also taken care to clarify those results.
> While we would ideally be able to provide an exhaustive comparison between our method and other policy optimization algorithms, such a comparison would likely be difficult to interpret within the scope of this work. Since all the algorithmic variations we consider are based on DDPG, directly comparing the results from each variant is straightforward. Along the same lines, reward relabeling is most straightforward in the application of off-policy algorithms. Figuring out the best way to perform adversarial reward relabeling for on-policy algorithms, such as PPO, is not something we can adequately address within the scope of our current work. However, to demonstrate that the challenge is not due simply to inadequacies of DDPG, we did attempt training with PPO as a baseline (without any relabeling). The results of PPO by itself are essentially the same as the results of DDPG by itself: it doesn’t work. The combination of sparse reward and difficult exploration are enough to make both baselines fail. To make this point available to the reader, we have included these results in Figure 1, where we also show vanilla DDPG.
> We thank the reviewer for pointing out the confusion in how we compare HER and CER. We have re-written portions of the paper to make this clearer. The point we wish to communicate is simply that HER and CER likely address different challenges within this domain of RL and are easily combined. Given that the two methods are more powerful in tandem, they do seem to interact in practice, so we agree that describing them as “orthogonal” may be confusing. We have included additional analyses to provide some intuition on how CER interacts with HER.

---

### Official Review · AnonReviewer2 · 2018-11-02

**Rating:** 7
**Confidence:** 4

**Review:**

The authors propose a new method for learning from sparse rewards in model-free reinforcement learning settings. This is a challenging and important problem in model-free RL, mainly due to the lack of effective exploration. They propose a new way of densifying the reward by encouraging a pair of agents to explore different states (using competitive self-play) while trying to learn the same task. One of the agents (A) receives a penalty for visiting states that the other agent (B) also visits, while B is rewarded for visiting states found by A. They evaluate their method on a few tasks with continuous action spaces such as ant navigation in a maze and object manipulation by a simulated robotic arm.  Their method shows faster convergence (in some cases) and better performance than comparable algorithms.


Strengths:
Attempts to solve a long-standing problem in model-free RL (effective exploration in sparse reward environments)
Clear writing and structure, easy to understand (except for some minor details)
Novel, intuitive, and simple method building on ideas from previous works
Good empirical results (better than state of the art, in terms of performance) on some challenging tasks

Weaknesses:
Not very clear why (and when) the method works -- more insight from experiments in less complex environments or some theoretical analysis would be helpful
It would also be useful to better understand the conditions under which we can expect this to bring significant gains and when we can expect this to fail (or not help more than other methods)
Not clear how stable (to train) and robust (to different environment dynamics) the method is


Main Comments / Questions:
The paper makes the claim that their technique “automatically generates a curriculum of exploration” which seems to be based more on intuition rather than clear experiments or analysis. I would suggest to either avoid making such claims or include stronger evidence for that. For example, you could consider visualizing the visited states by A and B (for a fixed goal and initial state) at different training epochs. Other such experiments and analysis would be very helpful.
It is known that certain reward shaping approaches can have negative consequences and lead to undesired behaviors (Ng et al., 1999; Clark & Amodei, 2016). Why can we expect that this particular type of reward shaping doesn’t have such side effects? Can it be the case that due to this adversarial reward structure, A learns a policy that takes it to some bad states from which it will be difficult to recover or that A & B get stuck in a cyclic behavior? Have you observed such behaviors in any of your experiments?
Do you train the agents with using the shaped reward (from the exploration competition between A and B) for the entire training duration? Have you tried to continue training from sparse reward only (e.g. after the effect ratio has stabilized)? One problem I see with this approach is the fact that you never directly optimize the true sparse reward of the tasks, so in the late stages of training your performance might suffer because the agent A is still trying to explore different parts of the state space.
Can you comment on how stable this method is to train (given its adversarial nature) and what potential tricks can help in practice (except for the discussion on batch size)?
Please make clear the way you are generating the result plots (i.e. is A evaluated on the full task with sparse reward and initial goal distribution with no relabelling?).
In Algorithm 1, can you include the initialization of the goals for A and B? Does B receive identical goals as A?
It would also be helpful to more clearly state the limitations and advantages of this method compared to other algorithms designed for more efficient exploration (e.g. the need for a resettable environment for int-CER but not for ind-CER etc.).


Minor Comments / Questions:
You might consider including more references in the Related Work section that initializing from different state distributions such as Hosu & Rebedea (2016), Zhu et al. (2016), and Kakade & Langford (2002), and perhaps more papers tackling the exploration problem.
Can you provide some intuition on why int-CER performs better than ind-CER (on most tasks) and why in Figure 1, HER + int-CER takes longer to converge than the other methods on the S maze?
In Figure 4, why are you not including ind-CER (without HER)?
Have you considered training a pool of agents with self-play (for the competitive exploration) instead of two agents? Is there any intuition on expecting one or the other to perform better?


Plots:
What is the x-axis of the plots? Number of samples, episodes, epochs? Please label it.
Please be explicit about the variance shown in the plots. Is that the std?
It would be helpful if to have larger numbers on the xy-axes. It is difficult to read when on paper.
Can you explain how you smoothed the curves -- whether before or after taking the average and perhaps include the min and max as well. I believe this could go in the Appendix.

Notation:
I don’t understand the need for calling the reward r_g instead of r. I believe this introduces confusion since the framework already has r taking as argument the goal g (eq. 1) while the g in the subscript doesn’t seem to refer to a particular g but rather to a general fact (that this is a reward for a goal-oriented task with sparse reward, where the goals are a subset of the states) (eq. 4)
Please use a consistent notation for Q. In sections 2.1 and 2.2, at times you use Q(s,a,g), Q(a,s,g) or Q(s,a).

Typos:
Page 6, last paragraph of section 4.1: Interestingly, even the … , is enough to support …
Page 7, last paragraph of section 4.3: Interestingly, … adversely affects both ...

---

> ### Author Response · Authors · 2018-11-25
> **Response to Reviewer 2**
>
> We would like to thank the reviewer for their very thoughtful, constructive, and encouraging feedback. Within the timeframe allowed for rebuttals, we have not had a chance to apply a more reductionist/theoretical approach. We hope to address such questions in future work.
> From our observations, CER is very stable if implemented correctly (as is HER by itself). This is quickly obvious when considering the variability in performance when using curiosity instead of CER (see new baselines in the revised paper). However, implementation matters. We found that our results were most consistent and strong when we periodically reset the parameters of agent B during the initial stages of training. In addition, we observe benefits in this regard from using a large batch size. We have made these details clearer in the revised paper, but found it necessary to include them in the appendix for space considerations.
> While HER helps the agent learn to reach arbitrary goals amongst the states it is capable of reaching, CER incentivizes the agent to encounter hard to reach states. The combination of these 2 strategies are powerful in the settings we have explored. However, one can imagine task/goal structures where generalizing from arbitrary goals is difficult to the point that CER does not help to explore useful directions. From a more technical perspective, CER may be difficult to apply when it is not straightforward to define reward based on the proximity of two substates.
> We have taken care to revise the figure annotations and notation to address the issues included in the review. In addition, we have included a new analysis based on the reviewer’s suggestion where we visualize the state distribution of agents trained on a U-maze during training.
> One likely strength of the reward shaping induced by CER comes from the fact that the only guaranteed reward is that related to the task itself. While the behavior of one agent could create an exploit for the other agent, we have never observed that to dominate learning such that policies oscillate between globally suboptimal behaviors. That said, it is possible that we have missed some examples of that kind of failure case and/or that the results might suffer if the balance of each reward type is off. From our empirical results, however, it does not seem that such degenerate behavior is an issue, and we did not have to carefully tune rewards.
> We have not tried scheduling CER until a certain point. Indeed, that may unmask some improvements if done correctly. However, it would require learning a new Q-function since we would effectively be switching to a single-player game with a new reward function. How to do that without inviting instability is not trivial and we did not get the chance to address that challenge.
> Following the reviewer’s suggestions, we have made it clearer when the two variations of CER are to be used and how that may affect the inherent limitations of CER.
> We did not experiment with a larger pool of agents. Using >2 agents while keeping the reward formulation the same (i.e. treating CER as a competition between 2 agents) would give the opportunity to relabel transitions using a more diverse set of agent B’s. Speculatively, richer sampling of competitor strategies could make CER more effective. However, this approach may exacerbate nonstationarities during learning. In any case, it is not immediately obvious where/how to modify CER to best support additional agents.

---

### Official Review · AnonReviewer4 · 2018-11-10
**Interesting idea; lack of comparisons with current methods.**

**Rating:** 5
**Confidence:** 4

**Review:**

The author proposes to use a competitive multi-agent setting for encouraging exploration.

I very much agree with most of previous reviewers, and their constructive suggestions. However, I find a major issue with this paper is the lack of baseline comparisons. The paper shows that CER + HER > HER ~ CER. I do not think CER should be compared to HER at all. CER to me attacks the exploration problem in a very different way than HER. It is not trying to "reuse" experience, which is the core in HER; instead, it uses 2 agents and their competition for encouraging visiting new states. This method should be compared to method that encourages exploration via some form of intrinsic motivation. There are methods proposed in the past, such as [1]/[2] that uses intrinsic motivation/curiosity driven prediction error to encourage exploration. Note that these methods are also compatible with HER. I'd suggest comparing CER with one of these methods (if not all) both with and without HER.

Minor:
In the beginning paragraph of 3.1, the paper states:
"
While the re-labelling strategy introduced by HER provides useful rewards for training a goal-conditioned
policy, it assumes that learning from arbitrary goals will generalize to the actual task goals. As such,
exploration remains a fundamental challenge for goal-directed RL with sparse reward. We propose a relabelling
strategy designed to overcome this challenge.
"
I think overcoming this particular challenge is a bit overstating. The method proposed in this paper is not guaranteed to address the "fundamental challenge" either --- i.e., why can you assume that learning from arbitrary goals that results from the dynamics of two agents will generalize to the actual task goals?

I will change my rating accordingly if there are more meaningful comparisons made in the rebuttal.

[1] Curiosity-driven Exploration by Self-supervised Prediction, Pathak et. al.
[2] Large-Scale Study of Curiosity-Driven Learning. Burda et. al.

---

> ### Author Response · Authors · 2018-11-25
> **Response to Reviewer 4**
>
> Thank you for your constructive review. The review has pointed out that, since CER should not be viewed as an alternative to HER, it is incomplete to treat HER as a suitable baseline. We agree with this assessment and have made sure to include results with HER+ICM (intrinsic curiosity module [1][2]). To compare CER with ICM, we discretized the action space into 5 bins per dimension for maze environments, and for robotic control environments we discretized the action space into 10 bins per dimension. We adapt ICM on top of HER for fair comparison. Experimental results show that CER is better than ICM in terms of faster learning and higher performance. Furthermore, ICM introduces a source of variability that CER manages to sidestep.
> Following the reviewer’s comments, we have selected more conservative language for motivating our method. We cannot provide a guarantee that our method allows the agent to generalize to task-relevant goals. Instead, we attempt to formulate a competition that incentivizes exploration and leverage the fact that the dynamics of this competition effectively yield an automatic curriculum of exploration. Our intuition is that, when used with HER, CER may help to discover arbitrary goals that more readily generalize to task-relevant goals.

---

> > ### Comment · AnonReviewer4 · 2018-12-09
> > **Please add details of HER + ICM**
> >
> > I'm glad to see that HER + CER was indeed better than HER + ICM, in terms of stability and faster learning.
> > However, I do not find any implementation details of HER + ICM. Could you describe how it was done and how did you tune the hyperparamaters for this method? Basically the question is how can I be convinced that the baseline is tuned properly -- as much as you have tuned for your method. Please include these details in your next revision of the paper.

---

> > > ### Author Response · Authors · 2018-12-13
> > > **Response to Reviewer 4**
> > >
> > > Thank you very much for your reply!
> > > The following are the necessary details to get HER+ICM to work on those tasks:
> > > We adopt code accompanying the paper 'Large-Scale Study of Curiosity-Driven Learning' to implement HER+ICM for fair comparison with HER+CER. An intrinsic reward is computed via ICM for each transition during sampling samples from environment, for each episode sampled from replay buffer HER is utilized to relabel and recompute extrinsic reward in each transition, then each transition has a weighted sum of intrinsic reward and extrinsic reward, finally we sample random transitions in these episodes for gradient computation. The weight between extrinsic reward and intrinsic reward is tuned between 0.1, 0.3, 0.6, 0.9 for each task.
> > > In order to get ICM to work on continuous control tasks, we discretize the action space into 5 bins per dimension for maze environment, and we discretize the action space into 10 bins per dimension for robotic control environment. We use 256x256 neural network to implement forward network and inverse network, and 128x128 neural network for embedding network, we use 3 layers neural network with hidden size 256 for policy and critic network, the same as in HER + CER. Buffer size, batchsize and number of parallel MPI cores are kept the same as HER + CER. To get HER + ICM to work on the four tasks, we additionally tune the ratio between HER replays and regular replays in from 2,3,4,5,6, and tune the learning rate of policy network and critic network and ICM between 0.001, 0.0003, 0.0001 for each task.
> > > We will add this detail in the experiment sections of the paper in next revision, and will be happy to add further clarifications if the reviewer requests it.

---

### Meta-Review · Area_Chair1 · 2018-12-14
**Nice idea with good empirical results, but ad hoc approach**

**Confidence:** 3
**Recommendation:** Accept (Poster)

**Metareview:**

The paper proposes a new method to improve exploration in sparse reward problems, by having two agents competing with each other to generate shaping reward that relies on how novel a newly visited state is.

The idea is nice and simple, and the results are promising.  The authors implemented more baselines suggested in initial reviews, which was also helpful.  On the other hand, the approach appears somewhat ad hoc.  It is not always clear why (and when) the method works, although some intuitions are given.  One reviewer gave a nice suggestion of obtaining further insights by running experiments in less complex environments.  Overall, this work is an interesting contribution.